# Transcriptomic and Non-Targeted Metabolomic Analyses Reveal Changes in Metabolic Networks during Leaf Coloration in *Cyclocarya paliurus* (Batalin) Iljinsk

Caowen Sun [1,2], Shengzuo Fang [1,2] and Xulan Shang [1,2,*]

1    College of Forestry, Nanjing Forestry University, Nanjing 210037, China; scw19871217@126.com (C.S.);
     fangsz@njfu.edu.cn (S.F.)
2    Co-Innovation Center for Sustainable Forestry in Southern China, Nanjing Forestry University,
     Nanjing 210037, China
*    Correspondence: shangxulan@njfu.edu.cn

**Abstract:** Secondary metabolites in *Cyclocarya paliurus* (Batalin) Iljinsk. leaves are beneficial for human health. The synthesis and accumulation of secondary metabolites form a complex process that is influenced by the trade-off between primary and secondary metabolism and by the biosynthetic pathways themselves. In this study, we explored the relationship between secondary metabolite accumulation and the activity of metabolic networks in leaves of *C. paliurus*. Leaves at three different growth stages were subjected to transcriptomic and non-targeted metabolomic analyses. The results revealed that nitrogen assimilation increased and carbon assimilation decreased as leaves matured, and the patterns of secondary metabolite accumulation and gene expression differed among the leaves at different growth stages. Mature green leaves had higher nitrogen assimilation and lower carbon assimilation, which were correlated with variations in secondary metabolite accumulation. As a major source of carbon and nitrogen, glutamine accumulated in the mature green leaves of *C. paliurus*. The accumulation of glutamine inhibited phenylalanine biosynthesis by modulating the pentose phosphate pathway but promoted acetyl-CoA biosynthesis through the tricarboxylic acid cycle. These changes led to decreased flavonoid contents and increased triterpenoid contents in mature leaves. These metabolomic and transcriptomic data reveal the differential expression of metabolic regulatory networks during three stages of leaf development and highlight the trade-off between primary and secondary metabolism. Our results provide a comprehensive picture of the metabolic pathways that are active in the leaves of *C. paliurus* at different growth stages.

**Keywords:** *Cyclocarya paliurus*; metabolic networks; transcriptome; secondary metabolites; leaf development





## 1. Introduction

*Cyclocarya paliurus* (Batalin) Iljinsk., a tree species found only in subtropical China, has leaves that contain health-promoting secondary metabolites, including flavonoids, triterpenoids, and phenolic acids. Thus, this tree species has a unique medicinal and health function and has great potential for commercial development [1–4]. Secondary metabolite-containing extracts from *C. paliurus* can improve insulin sensitivity, regulate disordered glucose and lipid metabolism, and reduce blood lipid contents and blood pressure [5]. However, the secondary metabolite components in *C. paliurus* leaves are variable and the bioactive function of the products is not uniform, which have hindered the development of the *C. paliurus* industry.

The pattern of secondary metabolite accumulation in *C. paliurus* leaves has been explored in previous studies. Such studies have discovered the key transcription factors and enzymes in the flavonoid and triterpenoid biosynthetic pathways, including MYBs and UDP-glucose: flavonoid 3-O-glucosyltransferases (UFGTs), as well as their regulation mechanisms [6–8]. The fertilizer level and ratio of nitrogen to carbon affect the balance between plant growth and defense, which in turn affect biomass and secondary metabolite

production [9,10]. Plant growth and the types and amounts of secondary metabolites also differ among genotypes and among populations growing in different geographical regions [11,12]. While most studies have focused on primary metabolism, secondary metabolism, or the relationship between the two, none has fully elucidated the metabolic network of *C. paliurus* and its relationship with secondary metabolite accumulation.

Plant secondary metabolite biosynthesis is affected by biotic and abiotic factors. The results of previous studies are not always consistent due to the complex effects of these factors on biosynthesis. Non-targeted metabolomic and transcriptomic analyses can provide descriptive data on secondary metabolite biosynthesis. In previous research, non-targeted metabolomic and transcriptomic methods have been used to study tea, with the ultimate goal of improving the leaf quality by regulating the contents of primary and secondary metabolites including flavonoids, amino acids, and polyphenols, and the ratio of polyphenols and amino acids [13–15]. Non-targeted metabolomic methods have also been used to analyze the development of fruit and leaf traits of other economically important plants [16–18]. In this study, transcriptomic and non-targeted metabolomic analyses were performed on leaves at different growth stages, with the aim of investigating the variability of the metabolic network of *C. paliurus*. This information will provide new insights into the secondary metabolism of *C. paliurus* and how it changes as leaves mature.

## 2. Material and Methods

### 2.1. Plant Materials

The plants used in this study were grown at the germplasm resource nursery for *C. paliurus*, which is located in Nanjing, China. This area is in a mid-subtropical region. The annual average temperature is 15.4 °C, the annual frost-free period is 237 days, the annual average sunshine index is 2240 h, and the annual average rainfall is 1087.4 mm. The laboratory experiments were conducted at Nanjing Forestry University in Xuanwu District of Nanjing, Jiangsu Province, China.

Three-year-old *C. paliurus* clonal seedlings were sampled. Leaf samples were collected at three stages: the young red-leaf stage, intermediate stage, and mature green-leaf stage. Three samples were collected for each stage, from plants with similar growth under homogeneous site conditions (Figure 1). Leaves with consistent age, similar growth, and no obvious diseases or pests were selected for analysis. The samples were frozen rapidly in liquid nitrogen, transferred to dry ice, and stored at −80 °C until metabolic and transcriptomic analyses.

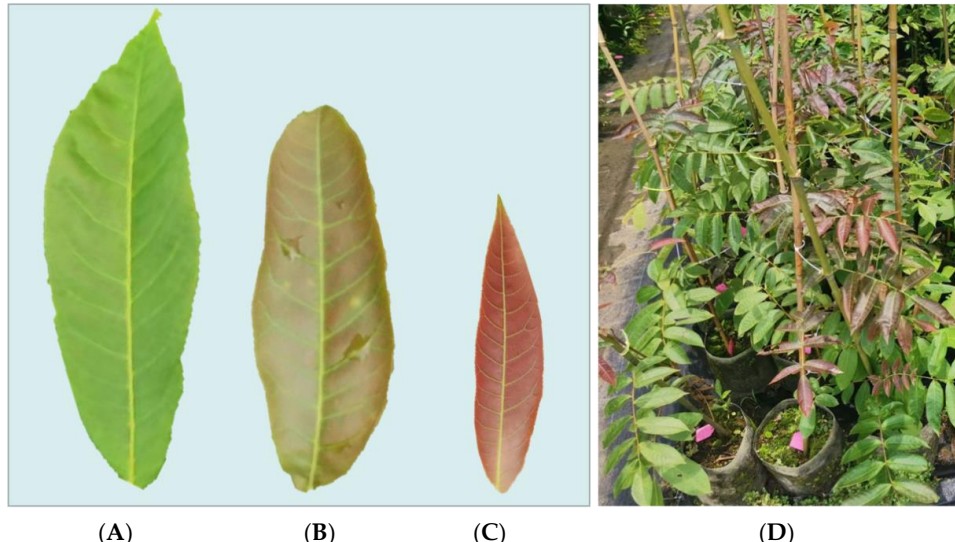

(**A**)  (**B**)  (**C**)  (**D**)

**Figure 1.** Different stages of leaf growth in clonal *C. paliurus* saplings with red leaves. (**A**) = mature green-leaf stage; (**B**) = intermediate stage; (**C**) = young red-leaf stage; (**D**) = *C. paliurus* saplings with leaves at different stages of growth.

### 2.2. Determination of Metabolites

Metabolomic analyses were conducted using ultra-high performance liquid chromatography (UPLC) and tandem mass spectrometry (MS/MS) (Applied Biosystems 4500 QTRAP). Metabolites were identified using an in-house database. The methods used for sample preparation and extraction, UPLC separation, MS/MS, and qualitative and quantitative determination of metabolites were as described by Sun et al. (2021) [19]. Wuhan MetWare Biotechnology Co., Ltd. (www.metware.cn, accessed on 10 May 2023) provided technical support for the metabolomics analysis.

### 2.3. Transcriptome Analysis

Transcriptome data were obtained for leaf samples at three different growth stages. The transcriptome data were mapped to the reference genome for tetraploid *C. paliurus*, which we determined in our previous research [6]. The reference genome data for tetraploid *C. paliurus* have been deposited in GenBank at the NCBI (https://www.ncbi.nlm.nih.gov/, accessed on 15 May 2023) under the accession number MW118603.1. The methods used for RNA extraction from samples, RNA quantification, and sequencing of leaf transcriptomes were as described previously [19]. Transcriptome sequencing was performed on the Illumina Hiseq2500 platform to generate 150-bp paired-end sequences, with technical support from the Wuhan Maivey Company. The raw data of RNA sequences generated in this study have been deposited at the NCBI (www.ncbi.nlm.nih.gov/bioproject/PRJNA723183, accessed on 17 May 2023) [20].

The transcriptomic data were saved in FastQC format and then assembled by Trinity (version r20140717) using the paired-end method. Functional annotations were assigned to genes using the Kyoto Encyclopedia of Genes and Genomes (KEGG), Gene Ontology (GO), and Clusters of Orthologous Groups of proteins (COG) databases. Gene transcript levels in the UniGene library were determined using the RSEM method with Bowtie. The number of reads mapped to each gene was determined using Feature Counts v1.5.0-p3, and then FPKM (exon fragments per million fragments) values were calculated. Differentially expressed genes (DEGs) were identified using the DESeq2R package in R. The DEGs were identified as those showing a >2-fold change in transcript levels in pairwise comparisons. The false discovery rate (FDR) correction for DEGs was set to $p < 0.05$. Enrichment analysis of DEGs was conducted using the KEGG and GO platforms.

### 2.4. Data Analysis

The DEGs and differential metabolites detected in the same comparison group were mapped to the KEGG pathway map to better understand the relationship between genes and metabolites.

According to the results of KEGG enrichment analysis of differential metabolites and DEGs, the KEGG pathways coenriched with both differential metabolites and DEGs were selected. Statistical analyses such as multiple comparisons, cluster analysis, and analysis of variance were performed using SPSS v. 17.0 software. Principal component analysis was performed using Rv.2.5.0 (R Development Core Team, 2007) [19]. Canonical correlation analysis (CCA) of the DEGs and differential metabolites in each selected pathway were conducted using the CCA package in R. Cluster analysis and principal component analysis of metabolome data were conducted and the cluster heat map was constructed using tools in R (www.r-project.org/, accessed on 30 June 2023) [19]. The fold change values and VIP values of the orthogonal partial least squares discriminant analysis (OPLS-DA) model were combined to screen for metabolites showing significant differences among the three leaf samples using the KEGG database. Differential metabolites were annotated and displayed and then subjected to KEGG classification and enrichment analysis based on pathway types.

## 3. Results

### 3.1. Metabolism and Transcriptome Association Analysis

In the PCA of the metabolomic data, the leaf samples formed three separate groups (Figure 2A), confirming that *C. paliurus* leaves at the three growing stages showed significantly different metabolite compositions. A heat map was constructed to visualize the upregulated and downregulated metabolites in leaves at different stages of development (Figure 3B). There were differences in the concentrations of phenolic acids, flavonoids, amino acids, nucleotides, and triterpenoids among leaves at different growth stages. Compared with mature green leaves (A stage), the young red leaves (C stage) had higher concentrations of phenolic acids, flavonoids, and some amino acids. However, the concentrations of triterpenoids and tannins were higher in mature green leaves than in young red leaves.

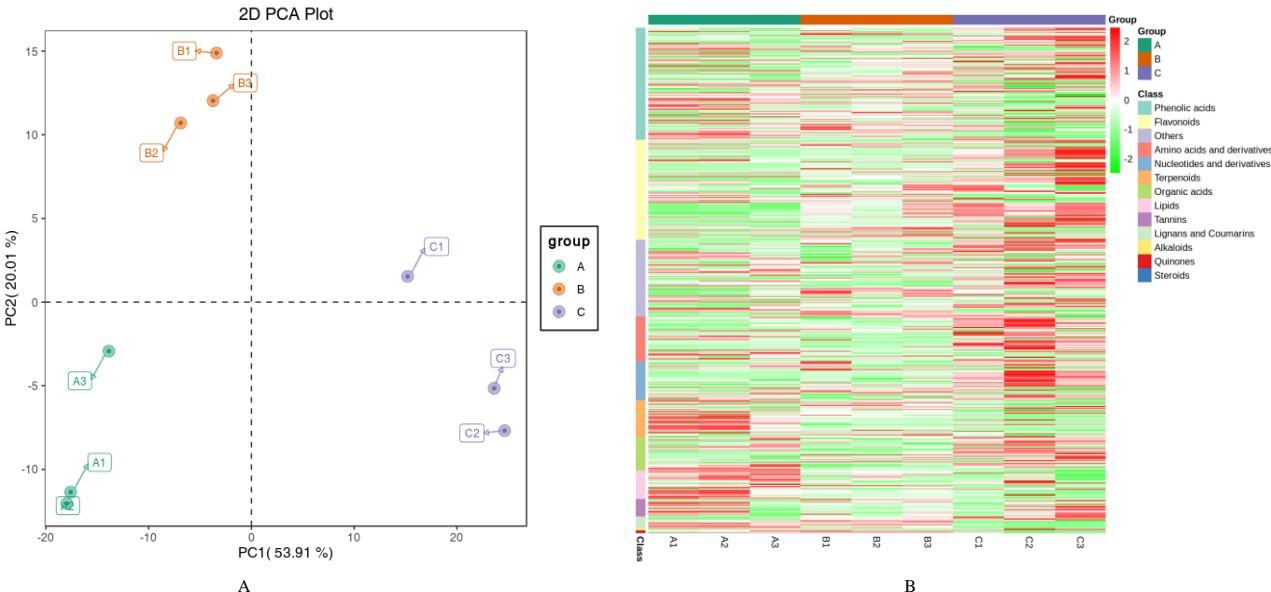

**Figure 2.** Metabolism analysis on *C. paliurus* (Batalin) Iljinsk. leaves at different growth stages. Note: (**A**), PCA plots showing metabolite accumulation patterns in *C. paliurus* leaves at different growth stages. A = mature green-leaf stage; B = intermediate stage; C = young red-leaf stage; (**B**), Heatmap clustering of three replicates of *C. paliurus* leaf samples at different stages based on metabolite profiles. Red indicates upregulated metabolites and green indicates downregulated metabolites. A = mature green-leaf stage; B = intermediate stage; C = young red-leaf stage.

According to the KEGG enrichment analyses (Figure 3A), the activities of amino acid biosynthesis, fatty acid biosynthesis, and flavone and flavonol biosynthesis pathways differed among the three stages of leaf growth. Our results revealed seven related DEGs in carbon metabolism, eight DEGs in phenylpropanoid biosynthesis, seven DEGs in flavonoid biosynthesis, and four DEGs in the pentose phosphate pathway. These results indicated that both primary and secondary metabolism differed among the three stages of leaf growth. According to the associated OPLS-DA analysis (Figure 3B), the metabolites most strongly correlated with DEGs were selected. These metabolites consisted of four flavonoids (quercetin-3-O-(6″-p-coumaroyl)galactoside, kaempferol-3-O-glucuronide-7-O-glucoside, 6-hydroxykaempferol-3,6-O-diglucoside, and catechin-(7,8-bc)-4β-(3,4-dihydroxyphenyl)-dihydro-2-(3H)-one); two phenolic acids (protocatechuic acid and 2,5-dihydroxybenzoic acid); one organic acid (2,3-dihydroxybenzoic acid); one tannin (gallic acid); and one other compound (N-acetyl-D-glucosamine-1-phosphate). These results showed that the biosynthesis of flavonoids and phenolic acids was regulated differently among leaves at different growth stages. To identify specific metabolites related to each biosynthesis pathway, we explored the relationships between genes and metabolites in different KEGG pathways.

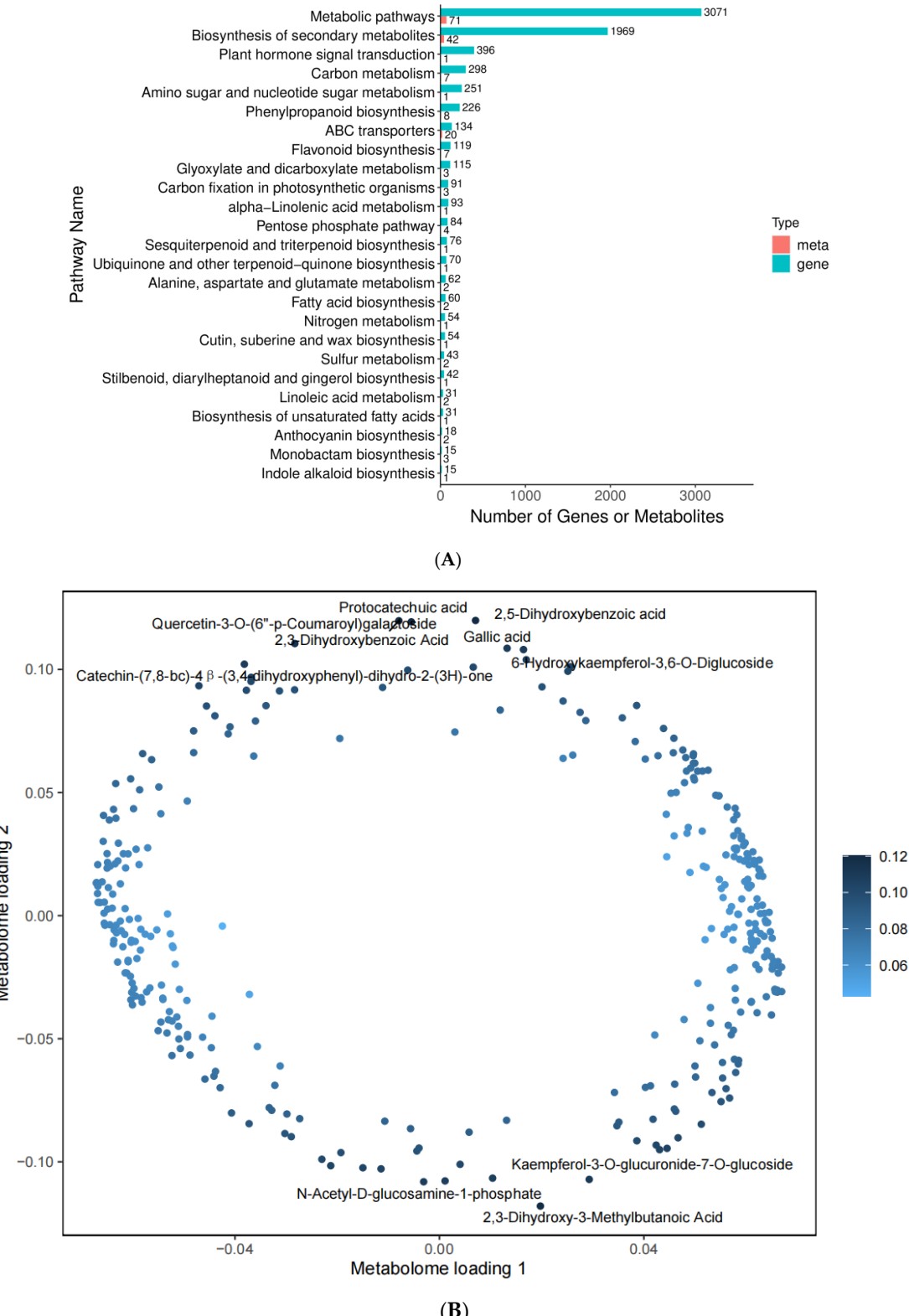

**Figure 3.** Metabolism and transcriptome association analysis in different growth stages of *C. paliurus* leaves. Note: (**A**) KEGG enrichment bar chart comparing mature green-leaf stage and young red-leaf stage. Horizontal coordinate represents the number of differential metabolites and differentially expressed genes enriched in the pathway, vertical coordinate shows KEGG pathway name. Red and green bars represent metabolome and transcriptome data, respectively; (**B**) Results of OPLS-DA analysis, showing metabolites most strongly correlated with DEGs.

### 3.2. N Metabolism

The results of the association analysis indicated that nitrogen metabolism significantly changed during leaf growth. As shown in Figure 4, genes encoding a nitrite transporter, nitrate reductase, and nitronate monooxygenase were significantly upregulated during leaf growth, indicating that nitrogen assimilation was enhanced as the leaves matured. As a result, the glutamine content was increased in mature green leaves, and the downregulation of glutamate synthase led to glutamine accumulation, which affected some related physiological activities (Table 1). Analyses of glutamine metabolism pathway enrichment revealed that both glutamine-fructose-6-phosphate transaminase and omega-amidase were upregulated, which supported the biosynthesis of D-glucosamine phosphate and 2-oxoglutarate (Figure 5). As a result, 2-oxoglutarate accumulated to higher levels as the leaves matured, thereby promoting the activity of the tricarboxylic acid (TCA) cycle. The upregulation of fructose 6-phosphate transaminase and glucose 6-phosphate isomerase during leaf development meant that D-glucosamine phosphate, which is a substrate for amino acid metabolism, was converted into beta-D-fructofuranose 6-phosphate and alpha-D-glucose 6-phosphate.

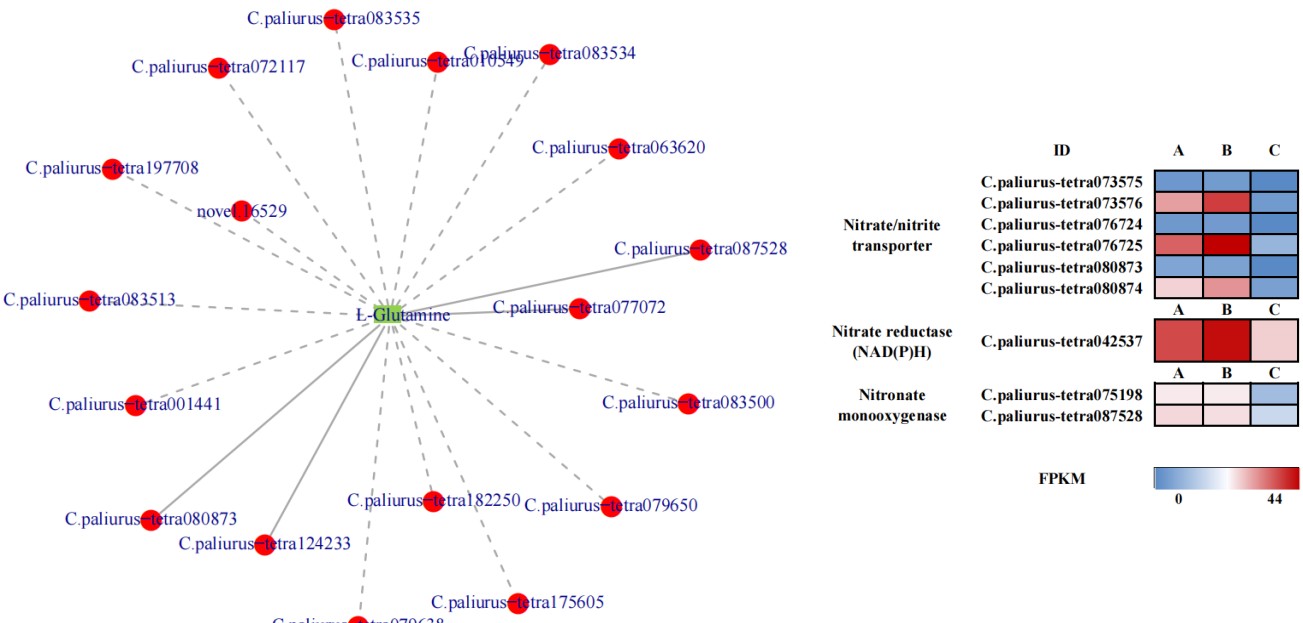

**Figure 4.** Correlation network diagram of nitrogen metabolism pathway. Green square indicates metabolite, red circles indicate genes. Solid lines indicate positive correlations and dashed lines indicate negative correlations.

According to the association analysis, glutamine accumulated to high concentrations in mature green leaves, and related genes also exhibited high transcript levels in mature green leaves. Based on the amino acid biosynthesis (Table 1), the increase in glutamine content was related to increased contents of lysine, tryptophan, histidine, N-acetyl-L glutamic acid, and ribulose-5-phosphate. Glutamine accumulation also led to reduced contents of D-erythrosine-4-phosphate, phenylalanine, arginine, and lutein, implying that nitrogen metabolism is regulated not only by amino acids but also by secondary metabolites.

**Table 1.** Differentially accumulated metabolites during *C. paliurus* (Batalin) Iljinsk. leaf growth and their related biosynthetic pathways.

| Metabolite | Stage | | | Class |
|---|---|---|---|---|
| | **A** | **B** | **C** | |
| L-Glutamine | $3.05 \times 10^6 \pm 1.45 \times 10^5$ b | $3.17 \times 10^6 \pm 2.03 \times 10^5$ b | $4.94 \times 10^6 \pm 6.38 \times 10^5$ a | Amino acids and derivatives |
| N-Acetyl-L-Glutamine | $3.05 \times 10^6 \pm 1.45 \times 10^5$ a | $3.17 \times 10^6 \pm 2.03 \times 10^5$ a | $4.94 \times 10^6 \pm 6.38 \times 10^5$ b | Amino acids and derivatives |
| L-Phenylalanine | $1.15 \times 10^7 \pm 7.09 \times 10^5$ a | $8.26 \times 10^6 \pm 5.12 \times 10^5$ a | $2.43 \times 10^7 \pm 4.36 \times 10^6$ b | Amino acids and derivatives |
| L-Aspartic Acid | $3.13 \times 10^5 \pm 1.55 \times 10^4$ a | $3.79 \times 10^5 \pm 3.03 \times 10^4$ a | $1.14 \times 10^6 \pm 2.67 \times 10^5$ b | Amino acids and derivatives |
| L-Lysine | $3.32 \times 10^5 \pm 1.00 \times 10^4$ b | $2.95 \times 10^5 \pm 8.33 \times 10^3$ b | $2.08 \times 10^5 \pm 1.82 \times 10^4$ a | Amino acids and derivatives |
| L-Histidine | $4.14 \times 10^4 \pm 2.10 \times 10^3$ b | $2.68 \times 10^4 \pm 2.65 \times 10^2$ a | $2.69 \times 10^4 \pm 3.61 \times 10^3$ a | Amino acids and derivatives |
| L-Arginine | $9.39 \times 10^4 \pm 3.73 \times 10^3$ a | $1.13 \times 10^5 \pm 5.69 \times 10^3$ a | $1.59 \times 10^6 \pm 5.41 \times 10^5$ b | Amino acids and derivatives |
| N-Acetyl-L-glutamic acid | $7.48 \times 10^4 \pm 9.68 \times 10^2$ c | $4.60 \times 10^4 \pm 2.67 \times 10^3$ b | $3.48 \times 10^4 \pm 4.51 \times 10^3$ a | Amino acids and derivatives |
| L-Tryptophan | $7.91 \times 10^5 \pm 1.08 \times 10^5$ b | $2.87 \times 10^5 \pm 2.29 \times 10^4$ a | $1.15 \times 10^5 \pm 2.60 \times 10^4$ a | Amino acids and derivatives |
| Syringin | $2.21 \times 10^5 \pm 2.19 \times 10^3$ b | $2.76 \times 10^5 \pm 8.62 \times 10^3$ c | $1.45 \times 10^5 \pm 1.15 \times 10^4$ a | Phenolic acids |
| Coniferaldehyde | $4.18 \times 10^4 \pm 3.20 \times 10^3$ b | $3.70 \times 10^4 \pm 3.70 \times 10^3$ b | $2.56 \times 10^4 \pm 9.81 \times 10^2$ a | Phenolic acids |
| Coniferyl alcohol | $3.44 \times 10^5 \pm 6.41 \times 10^4$ b | $3.33 \times 10^5 \pm 1.88 \times 10^4$ b | $1.33 \times 10^5 \pm 4.04 \times 10^3$ a | Phenolic acids |
| Cyanidin-3-O-glucoside | $2.87 \times 10^5 \pm 7.56 \times 10^4$ a | $3.08 \times 10^6 \pm 2.48 \times 10^5$ b | $8.07 \times 10^6 \pm 6.13 \times 10^5$ c | Anthocyanins |
| Delphinidin-3-O-glucoside | $2.11 \times 10^6 \pm 5.92 \times 10^5$ a | $1.87 \times 10^7 \pm 2.81 \times 10^6$ b | $8.85 \times 10^7 \pm 6.02 \times 10^6$ c | Anthocyanins |
| Luteolin | $9.30 \times 10^3 \pm 2.48 \times 10^3$ a | $1.24 \times 10^4 \pm 1.86 \times 10^2$ a | $6.54 \times 10^4 \pm 1.51 \times 10^4$ b | Flavonoid |
| Kaempferol-3-O-rutinoside | $1.20 \times 10^5 \pm 3.51 \times 10^3$ c | $7.18 \times 10^4 \pm 1.61 \times 10^4$ b | $4.28 \times 10^3 \pm 4.77 \times 10^2$ a | Flavonoid |
| Quercetin | $3.31 \times 10^5 \pm 4.80 \times 10^4$ a | $5.06 \times 10^5 \pm 2.80 \times 10^4$ a | $9.68 \times 10^5 \pm 1.37 \times 10^5$ b | Flavonols |
| Quercetin-3-O-xyloside (Reynoutrin) | $3.70 \times 10^6 \pm 1.76 \times 10^5$ a | $9.12 \times 10^6 \pm 5.47 \times 10^5$ b | $1.27 \times 10^7 \pm 1.56 \times 10^6$ c | Flavonols |
| Quercetin-3-O-glucoside (Isoquercitrin) | $7.87 \times 10^6 \pm 5.29 \times 10^5$ a | $9.95 \times 10^6 \pm 6.76 \times 10^5$ a b | $1.20 \times 10^7 \pm 7.75 \times 10^5$ b | Flavonols |
| Quercetin-3-O-rutinoside (Rutin) | $2.62 \times 10^4 \pm 5.27 \times 10^3$ a | $3.44 \times 10^4 \pm 4.00 \times 10^3$ a | $1.70 \times 10^5 \pm 5.18 \times 10^4$ b | Flavonols |
| Kaempferol-3,7-O-dirhamnoside (Kaempferitrin) | $1.71 \times 10^5 \pm 3.60 \times 10^4$ a | $1.58 \times 10^5 \pm 4.98 \times 10^3$ a | $1.98 \times 10^5 \pm 6.18 \times 10^4$ a | Flavonols |
| Apigenin-6-C-glucoside (Isovitexin) | $4.88 \times 10^5 \pm 9.50 \times 10^3$ b | $1.69 \times 10^5 \pm 4.53 \times 10^4$ a | $1.30 \times 10^5 \pm 3.20 \times 10^4$ a | Flavonoid carbonoside |
| D-Erythrose-4-phosphate | $2.71 \times 10^4 \pm 2.11 \times 10^3$ a | $2.67 \times 10^4 \pm 2.43 \times 10^3$ a | $5.10 \times 10^4 \pm 8.61 \times 10^3$ b | Saccharides and Alcohols |
| Ribulose-5-phosphate | $3.20 \times 10^4 \pm 4.16 \times 10^2$ b | $1.38 \times 10^4 \pm 1.18 \times 10^3$ a | $1.89 \times 10^4 \pm 3.21 \times 10^3$ a | Saccharides and Alcohols |
| Shikimic acid | $1.06 \times 10^5 \pm 5.09 \times 10^3$ a | $1.01 \times 10^5 \pm 1.43 \times 10^4$ a | $2.21 \times 10^5 \pm 2.14 \times 10^4$ b | Organic acids |

Significant differences among the leaf growth stages are indicated by different lowercase letters ($p < 0.05$).

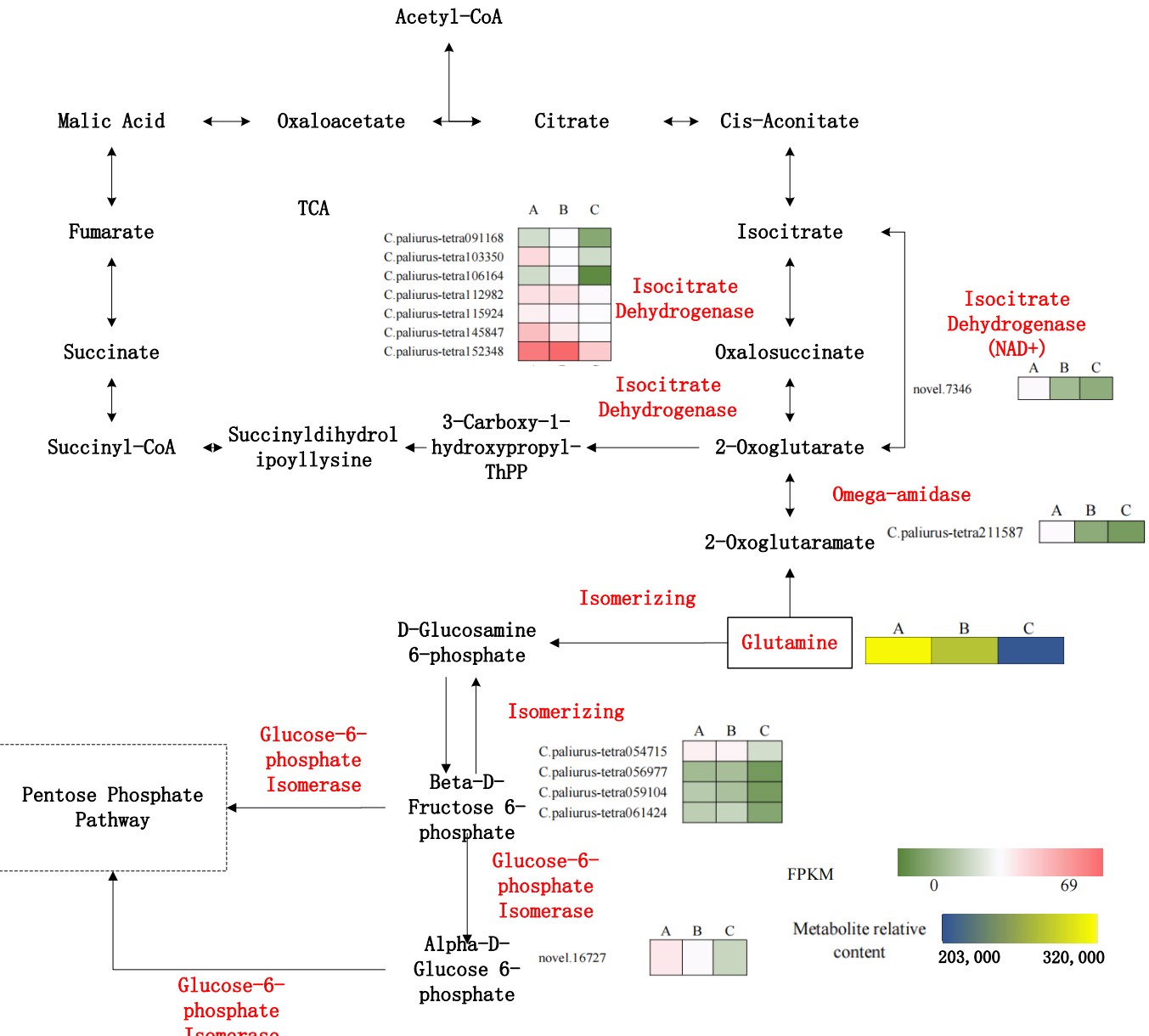

**Figure 5.** Modulation of genes related to nitrogen assimilation during leaf growth of *C. paliurus*. Histograms show FPKM values of genes and relative contents of metabolites. Charts in green and red represent down- and upregulated genes, respectively; graphs in blue and yellow represent down- and upregulated metabolites, respectively.

### 3.3. Relationships between Metabolites in Specific Pathways and DEGs

We conducted CCAs to explore the relationships between metabolites in specific pathways and up- and downregulation of gene expression during leaf development.

The CCA of the carbon fixation pathway showed that it was downregulated during leaf growth, and related genes in this pathway were highly correlated with the downregulation of D-erythrose-4-phosphate and aspartic acid (Figure 6A). These results highlight that nitrogen assimilation was upregulated and carbon assimilation was downregulated in mature green leaves compared with young red leaves. This may have affected downstream metabolic pathways, including secondary metabolism. As shown in Table 1, amino sugar metabolism provided substrates for ribose 5-phosphate biosynthesis in the growing leaves, while D-erythrose 4-phosphate accumulation was restrained by the downregulation of transketolase in the pentose phosphate pathway.

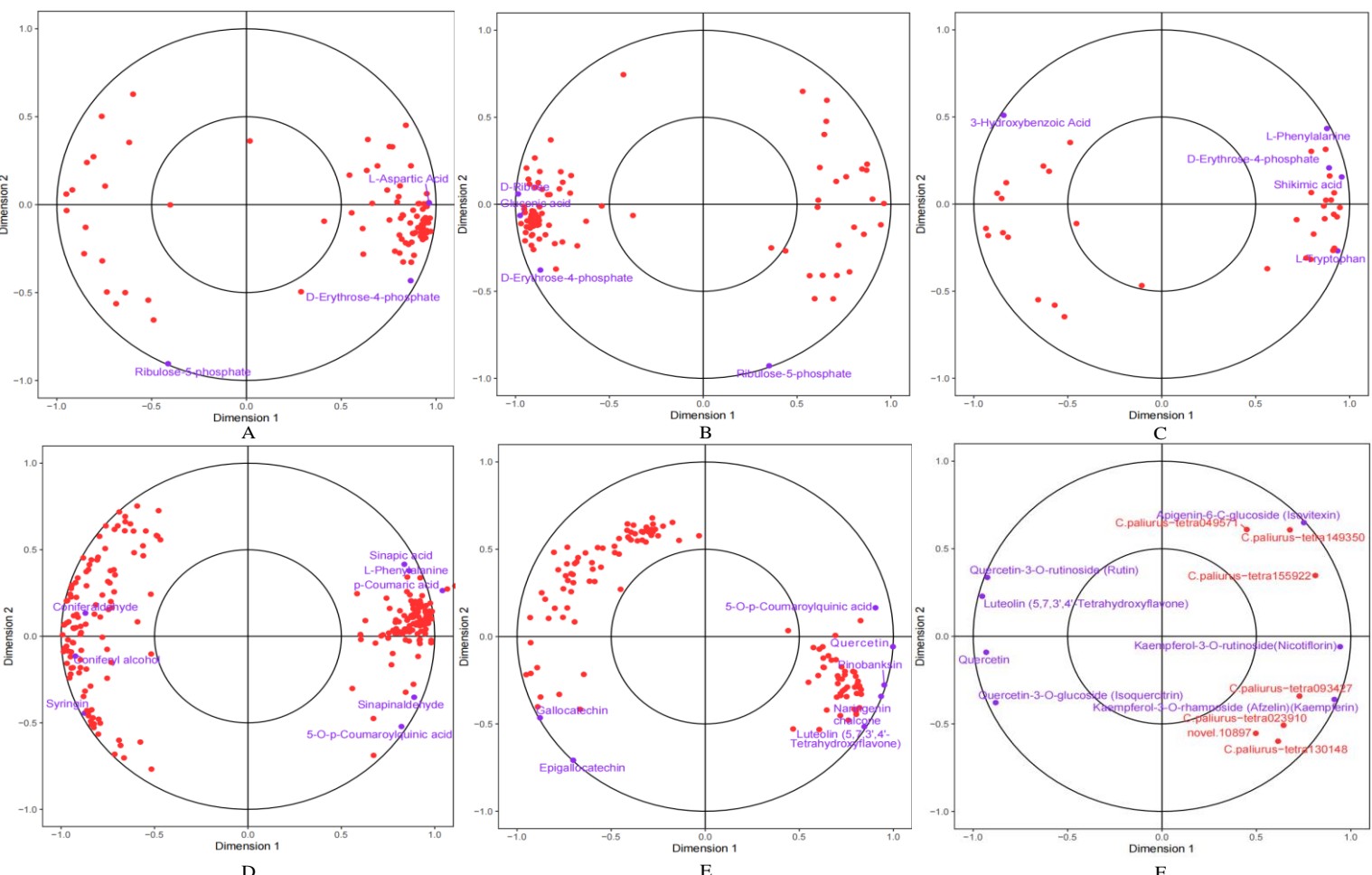

**Figure 6.** CCA of DEGs and differential metabolites in the KEGG enrichment pathways. Note: Metabolites are labeled in purple circles and genes in red circles. (**A**) CCA of DEGs and differential metabolites in the carbon fixation pathway; (**B**) CCA of DEGs and differential metabolites in the pentose phosphate pathway; (**C**) CCA of DEGs and differential metabolites in phenylalanine, tyrosine and tryptophan biosynthesis; (**D**) CCA of DEGs and differential metabolites in phenylpropanoid biosynthesis; (**E**) CCA of DEGs and differential metabolites in flavonoid biosynthesis; (**F**) CCA of DEGs and differential metabolites in flavone and flavonol biosynthesis.

In the CCA of the pentose phosphate pathway (Figure 6B), decreases in the contents of D-erythrose-4-phosphate, D-ribose, and gluconic acid were highly correlated with the expression levels of related genes. However, the increased ribulose-5-phosphate content was only weakly correlated with increased expression of related genes. The increase in ribulose-5-phosphate content might be attributed to the decrease in D-erythrose-4-phosphate content and limited availability of substrates for the downstream biosynthesis pathway (Table 1).

In the CCA of phenylalanine, tyrosine, and tryptophan biosynthesis (Figure 6C), phenylalanine, shikimic acid and tryptophan all decreased alongside decreased D-erythrose-4-phosphate content, and the related genes were also downregulated. The decreased D-erythrosine-4-phosphate content led to the accumulation of phenylalanine, shikimic acid, and tryptophan.

In the CCA of the shikimic acid pathway (Figure 6D), the decreased D-erythrose 4-phosphate content led to substrate limitation for phenylalanine metabolism. In the mature green leaf period, 3-deoxy-7-phospholipid synthase, chorismate kinase, and chorismate mutase were significantly downregulated because of the lower contents of shikimate and phenylalanine.

In the CCA of the phenylpropanoid biosynthesis pathway (Figure 6E), the decreased phenylalanine content was related to the decreased 5-O-p-coumaroylquinic acid content, and the related genes were also downregulated. Intermediate products of lignans, including coniferal dehyde, coniferal alcohol, and syringin accumulated continuously in the mature green-leaf stage. Mature green leaves tended to accumulate lignans despite substrate limitation, but flavonoid accumulation was limited as indicated by the reduced contents of naringenin chalcone, pinobanksin, luteolin, quercetin, and 5-O-p-coumaroylquinic acid.

In the flavonoid biosynthetic pathway, the naringenin chalcone content decreased significantly along with decreases in the contents of phenylalanine and 5-O-p-coumaroylquinic acid (Figure 6F). As the naringenin chalcone content decreased, the contents of gallocatechin and epigallocatechin increased during leaf growth. The increased contents of these compounds downstream of the flavonoid biosynthesis pathway, as well as the upregulation of genes related to flavonoid biosynthesis, indicates that some branches of the flavonoid biosynthesis pathway were upregulated despite the shortage of substrates.

In the flavone and flavonol biosynthesis pathway (Figure 6F), the contents of quercetin, quercetin-3-O-glucoside, and quercetin-3-O-rutinoside content decreased as the luteolin content decreased during leaf growth, whereas the contents of kaempferol-3-O-rhamnoside, apigenin-6-C-glucoside, and kaempferol-3-O-rutinoside increased during leaf growth. Despite the shortage of substrates, genes involved in flavone and flavonol biosynthesis were upregulated during leaf growth, leading to the accumulation of flavones. The biosynthesis of anthocyanins downstream of flavonol biosynthesis was inevitably limited by the lack of substrates, and the production of cyanidin-3-O-glucoside and delphinidin-3-O-glucoside decreased as the leaves matured.

### 3.4. Acetyl-CoA Biosynthesis

According to the transcriptome analysis, glutamine accumulation in the mature green leaves regulated not only the pentose phosphate pathway but also the TCA cycle (Figure 5). The regulation of gene expression related to omega amidase is known to affect the amount of 2-oxoglutarate entering the TCA cycle. Isocitrate dehydrogenase and $NAD^+$ were also enhanced facilitating the conversion of 2-oxoglutarate into isocitrate. This would facilitate the conversion of isocitrate to citrate, thereby providing more acetyl-CoA as the substrate for downstream pathways. Both acetyl-CoA and acetoacetyl-CoA are used as substrates by hydroxymethylglutaryl-CoA synthase, hydroxymethylglutaryl-CoA reductase (Figure 5), phosphomevalonate kinase, and diphosphomevalonate decarboxylase in the mevalonate pathway. This pathway synthesizes the terpenoid backbone, and its up-regulation leads to triterpenoid accumulation [19]. These findings indicate that the triterpenoid pathway was enhanced during leaf growth and that this pathway obtained more substrate (as acetyl-CoA) from nitrogen assimilation than from carbon assimilation.

## 4. Discussion

The biosynthesis and accumulation of secondary metabolites form a complex process that is largely influenced by the trade-off between primary and secondary metabolism, in addition to being regulated by biosynthetic pathways themselves. Thus, only limited conclusions can be drawn from studies on secondary metabolic pathways alone. Studies using a combination of metabolome and transcriptome analyses can reveal the differential expression of metabolic regulatory networks across different stages of leaf development. Characterization of the transcriptomic and metabolic profiles can clarify the trade-off between primary and secondary metabolism and comprehensively reflect the overall picture of metabolic pathway expression, thereby shedding light on the mechanisms by which different secondary metabolites accumulate during leaf growth.

Ruan et al. (2010) found that more glutamine accumulated in *Camellia sinensis* (L.) O. Kuntze tissues when they were supplied with sufficient nitrogen [13]. Similarly, we found that mature green leaves accumulated more glutamine than did young red leaves, suggesting that mature green leaves exhibit stronger nitrogen assimilation. Ruan et al. (2010) detected a small change in glutamate content and attributed it to homeostatic adjustment in the plant [13]. We also detected a small change in glutamate content in the maturing *C. paliurus* leaves in this study. The downregulation of glutamate synthase was related to the increase in glutamine content, which in turn restricted the biosynthesis of arginine and ornithine. Our results suggest that glutamine is a major source of carbon and nitrogen in mature green leaves, and its consumption may lead to carbon limitation. Lyubetskaya et al. (2006) suggested that carbon limitation adjusts the activity of the pentose phosphate pathway, with an increase in glucose-6-phosphate biosynthesis leading to ribulose-5-phosphate synthesis [21]. Consistent with the results of previous studies, we found that mature green leaves of *C. paliurus* tended to favor ribose-5-phosphate synthesis over D-erythrose-4-phosphate synthesis through enhanced glucose-6-phosphate synthesis, which further reduced the amount of substrate available for the phenylalanine biosynthesis pathway.

Flavonoid synthesis is mainly affected by phenylalanine ammonia-lyase. Qin et al. (2022) [10] also found that an intermediate nitrogen fertilizer level increased the contents of secondary metabolites, including flavonoids, triterpenoids, and polyphenols, in *C. paliurus* leaves. Sheng et al. (2021) found that the total flavonoid content first increased and then decreased during the growth of *C. paliurus* leaves, and attributed this to differences in the expression levels of genes encoding MYBs, UFGTs, and enzymes in the flavonoid biosynthesis pathway [22]. Du et al. (2022) observed that the total amino acid content continuously decreased during the development of *C. paliurus* leaves [23]. Yue et al. (2023) also found that the addition of nitrogen improved the photosynthetic efficiency of *C. paliurus* seedlings [24]. In contrast, Deng et al. (2019) found that carbon assimilation played a role in flavonoid accumulation, but an adequate nitrogen supply was also necessary [9]. The results of those studies implied that amino acids and secondary metabolite biosynthesis are connected within a metabolic network, but did not reveal details of their interactions. In the present study, we found that nitrogen restricted flavonoid accumulation via phenylalanine deficiency, rather than via downregulation of phenylalanine ammonia-lyase. Therefore, the decrease in flavonoid content during the growth of *C. paliurus* leaves may be attributed to the preferential supply of nitrogen to amino acid biosynthesis pathways, and to the limited supply of substrates for the flavonoid biosynthesis pathway.

In this study, we found that nitrogen strongly affected both primary and secondary metabolism in the leaves. Deng et al. (2019) [9] also considered that nitrogen availability significantly influenced flavonoid accumulation in *C. paliurus,* with a moderate level (0.63 mmol $NH_4NO_3$) of nitrogen fertilizer resulting in the highest total flavonoid content per plant [9]. Qin et al. (2021) [25] found that the nitrogen form significantly affected the accumulation of phenolic compounds. Deng et al. (2019) observed that the accumulation patterns of quercetin, isoquercitrin, and kaempferol were the same because they share the same biosynthesis pathway, and that the biosynthesis of flavanone-3-hydroxylase was strongly correlated with the total flavonoid content per plant, indicating that flavanone-3-

hydroxylase was the key regulating enzyme [9]. In the present study, however, quercetin and kaempferol showed different patterns of accumulation, despite the overall enhancement of the flavonoid biosynthesis pathway during leaf development. It may therefore be inferred that the lack of phenylalanine caused a decrease in the accumulation of flavonoids and changes in the composition of flavonoids during leaf growth. This means that the leaves of *C. paliurus* exhibited differences in nitrogen and carbon assimilation efficiency at different stages of development, further affecting secondary metabolite biosynthesis downstream.

Our results show that in the maturing leaves of *C. paliurus*, the phenylalanine biosynthesis pathway, which is upstream of flavonoid biosynthesis, was downregulated by the lack of D-erythrose 4-phosphate as the substrate. The lack of substrate limited the accumulation of most flavonoids, including cyanidin-3-O-glucoside and delphinidin-3-O-glucoside. The activity of associated enzymes in the flavonoid biosynthesis pathway was more active in older leaves than in younger ones, resulting in the continuous accumulation of kaempferol-3-O-rutinoside, a product of the flavonoid biosynthesis pathway. Triterpenoid synthesis increased in mature green leaves because of the regulation of the flux of the terpenoid backbone, for which acetyl-CoA serves as the primary substrate. Previously, we found that key structural genes involved in the synthesis of the triterpenoid skeleton showed significant differences in their expression levels among different stages of leaf development. Furthermore, the changes in gene expression were consistent with the accumulation patterns of most triterpenoid monomers during leaf growth. In addition, genes encoding glycosyltransferases were found to be upregulated as the leaves matured [19]. Our previous study also found that phosphomevalonate kinase and diphosphomevalonate decarboxylase in the mevalonate pathway of terpenoid backbone biosynthesis were upregulated, resulting in further triterpenoid accumulation [19]. The results of the present study are consistent with the idea that the utilization of glutamine by the TCA cycle promotes acetyl-CoA biosynthesis, which supplies sufficient substrate for the mevalonate pathway. In their study on *Ganoderma lucidum* cells, Lian et al. (2021) also found that the carbon skeletons integrated into triterpenoids came from the TCA cycle and glycolysis under nitrogen-limited conditions [26].

This study presents a comprehensive analysis of the differential expression of metabolic regulatory networks across different leaf growth stages in *C. paliurus*, based on integrated metabolome and transcriptome data. Furthermore, we elucidate the intricate trade-off between primary and secondary metabolism, providing an overall depiction of metabolic pathways. To further investigate this trade-off, validation of RNA-Seq results through qPCR is required for specific biosynthesis pathways.

## 5. Conclusions

Transcriptomic and non-targeted metabolomic analyses were performed for *C. paliurus* leaves at different stages of growth. The major physiological changes during leaf growth were identified. The mature green leaves exhibited higher nitrogen assimilation and lower carbon assimilation than the young red leaves of *C. paliurus*, and this was identified as the main cause of the metabolic changes in the leaves. As a major source of carbon and nitrogen, glutamine accumulated in the leaves of *C. paliurus* and inhibited phenylalanine biosynthesis by modulating the pentose phosphate pathway. However, the accumulated glutamine also promoted acetyl-CoA biosynthesis through the TCA cycle, leading to a decrease in flavonoid content and an increase in triterpenoid content. These findings provide new insights into secondary metabolite biosynthesis in *C. paliurus*.

**Author Contributions:** Methodology and experimental design, C.S.; data curation, C.S.; data analysis, C.S.; Prepared with original draft by C.S.; Written, reviewed and edited by S.F. and X.S.; Acquisition of funds, C.S., S.F. and X.S. All authors have read and agreed to the published version of the manuscript.

**Funding:** This work was supported by the Key Research and Development Program of Jiangsu Province (BE2019388), the National Natural Science Foundation of China (32201541), the Priority

Academic Program Development of Jiangsu Higher Education Institutions (PAPD) and the Doctorate Fellowship Foundation of Nanjing Forestry University. Funders had no role in the study. Design, data collection and analysis, decision to publish or preparation of the manuscript.

**Data Availability Statement:** The datasets generated during and/or analyzed during the current study are not publicly available due the data also form part of an ongoing study but are available from the corresponding author on reasonable request.

**Conflicts of Interest:** The authors declare no conflict of interest.

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
