# Peer review of "Transcriptomic and Non-Targeted Metabolomic Analyses Reveal Changes in Metabolic Networks during Leaf Coloration in Cyclocarya paliurus (Batalin) Iljinsk"

_forests, doi:10.3390/f14101948_

Round 1
Reviewer 1 Report
In the manuscript entitled “Transcriptomic and non-targeted metabolomic analyses reaveled differences in the metabolic networks during leaf coloring of Cyclocarya paliurus”, the authors showed the leaves of different growing stages of the C. paliurus clone with young red leaves were analyzed using transcriptomic and non-targeted metabolomic analyses.
The manuscript written comprehensively and the theme of the study is presented appropriately.
However, the authors should show validation of RNA-Seq results by qPCR. Also, references should be written according to the ‘forests’ journal format.
Author Response
Dear editor and reviewer,
I have revised all the tips suggested by reviewer. I want to show my thankful for your patience and work. Looking for good news and wish you a happy day.
Sun Caowen
In the manuscript entitled “Transcriptomic and non-targeted metabolomic analyses reaveled differences in the metabolic networks during leaf coloring of Cyclocarya paliurus”, the authors showed the leaves of different growing stages of the C. paliurus clone with young red leaves were analyzed using transcriptomic and non-targeted metabolomic analyses.
The manuscript written comprehensively and the theme of the study is presented appropriately.
However, the authors should show validation of RNA-Seq results by qPCR. Also, references should be written according to the ‘forests’ journal format.
Thanks, English language has been revised.
Thanks for your suggestion, it has been modified in the discussion “In total, this study presents a comprehensive analysis of the differential expression of metabolic regulatory networks across different leaf growth stages in C.paliurus, based on integrated metabolome and transcriptome data. Furthermore, we elucidate the intricate trade-off between primary and secondary metabolism, providing an overall depiction of metabolic pathways. To further investigate this trade-off, validation of RNA-Seq results through qPCR is required for specific biosynthesis pathways.”
Thanks, references has been written according to the ‘forests’ journal format.

Reviewer 2 Report
The authors analyzed the differential accumulation metabolites of Cylocarya paliurus leanves among three different stages, and screened the differential expression genes in N metabolism using the published RNA-seq data. The content of the manuscript could not reflect the title of “metabolic networks”. The results helped to understand the accumulation patterns of metabolites and transcriptional regulation basis in the leaf development of C. paliurus.
1. The last sentence in abstract said: this finding will help to better understand the accumulation patterns of metabolites and secondary metabolites in the leaves of C. paliurus at different growth stages. But I also could not understand the accumulation patterns and related genes expression patterns of the leaves at different stages. Therefore, the scientific questions are not put forward well, and the research conclusions are not summarized clearly in the abstract. My suggestion is that major changes should be made to the abstract.
2. Line32-33, the metabolite content is variable or the components is variable?
3. Line67, how old of the seedlings?
4. The method of sample and the set of duplicate should be more clearly. And “strains” is not applicable in this site.
5. The section of data analysis is too complex to understand. my suggestion is that the data analysis of metabolomics and transcriptome would be separated, then provide the method of joint analysis of differential expression genes and differential accumulation metabolites.
6. OPLS-DA model was used to select differential accumulation metabolites, but the result displayed O2PLS analysis.
7. Why focus on N metabolism? I could not understand. What is the relationship between N metabolism and the secondary metabolites of the research target you displayed in line 58.
8. Where is figure 9?
9. The note of picture in figure 8 is not clear, it is hard to distinct the genes and metabolites.
10. The CCA analysis between pathway genes and metabolites in biosynthesis pathway is not necessary. If it is determined that the pathway gene encoding the enzyme in the pathway of metabolite biosynthesis, then the up-regulated expression of this pathway gene will inevitably lead to an increase in the metabolite content.
Author Response
Dear editor and reviewer,
I have revised all the tips suggested by reviewer. I want to show my thankful for your patience and work. Looking for good news and wish you a happy day.
Sun Caowen
The authors analyzed the differential accumulation metabolites of Cylocarya paliurus leanves among three different stages, and screened the differential expression genes in N metabolism using the published RNA-seq data. The content of the manuscript could not reflect the title of “metabolic networks”. The results helped to understand the accumulation patterns of metabolites and transcriptional regulation basis in the leaf development of C. paliurus.
- The last sentence in abstract said: this finding will help to better understand the accumulation patterns of metabolites and secondary metabolites in the leaves of C. paliurus at different growth stages. But I also could not understand the accumulation patterns and related genes expression patterns of the leaves at different stages. Therefore, the scientific questions are not put forward well, and the research conclusions are not summarized clearly in the abstract. My suggestion is that major changes should be made to the abstract.
Thanks, the abstract has been rewritten.
- Line32-33, the metabolite content is variable or the components is variable?
Thanks, it was “the secondary metabolite components” and has been modified.
- Line67, how old of the seedlings?
Thanks, it was “3-year-old” and has been modified.
- The method of sample and the set of duplicate should be more clearly. And “strains” is not applicable in this site.
Thanks, it was modified as “Three-year-old C. paliurus clonal seedlings were sampled. Leaf samples were collected at three stages; the young red leaf stage, intermediate stage, and mature green leaf stage.”.
- The section of data analysis is too complex to understand. my suggestion is that the data analysis of metabolomics and transcriptome would be separated, then provide the method of joint analysis of differential expression genes and differential accumulation metabolites.
Thanks, the section of data analysis has been modified, and the data analysis of metabolomics and association analysis was separated including figures.
- OPLS-DA model was used to select differential accumulation metabolites, but the result displayed O2PLS analysis.
Thanks, the figures captions have been modified appropriately.
7.Why focus on N metabolism? I could not understand. What is the relationship between N metabolism and the secondary metabolites of the research target you displayed in line 58.
Thank you for your suggestion. The manuscript has been revised to enhance its comprehensibility. The study primarily focused on metabolic networks, including N metabolism, the carbon fixation pathway, the pentose phosphate pathway, phenylalanine biosynthesis, the shikimic acid pathway, the phenylpropanoid biosynthesis pathway, the flavonoid biosynthetic pathway, Acetyl-CoA biosynthesis and TCA cycle. These biosynthetic pathways exhibited a strong correlation with flavonoids and triterpenoids accumulation in C.paliurus. Furthermore, we discovered that N metabolism plays a crucial role in substrate accumulation and thus emphasized this pathway using a schematic diagram. Additionally, CCA analysis was conducted to investigate the regulation of other KEGG pathways.
- Where is figure 9?
Thanks, the figures have been modified.
- The note of picture in figure 8 is not clear, it is hard to distinct the genes and metabolites.
Thanks, the figures have been modified.
- The CCA analysis between pathway genes and metabolites in biosynthesis pathway is not necessary. If it is determined that the pathway gene encoding the enzyme in the pathway of metabolite biosynthesis, then the up-regulated expression of this pathway gene will inevitably lead to an increase in the metabolite content.
Thank you for your suggestion. The CCA analysis has been simplified, and the manuscript of this section has also been adjusted to serve as a supplement for illustrating the regulation of metabolic networks in C.paliurus leaves. The simplified CCA focuses on elucidating the connection between upstream and downstream biosynthesis pathways, aiming to provide a comprehensive description of the metabolic network.

Reviewer 3 Report
Title must be “Transcriptomic and non-targeted metabolomic analyses revealed differences in the metabolic networks during leaf coloring of Cyclocarya paliurus”
Synthetic accumulation of secondary metabolites is a complex process that is largely influenced by the trade-off between primary and secondary metabolism, in addition to being regulated by the synthesis pathway itself. This study successfully demonstrated roles of improved N- assimilation and lower carbon assimilation and the modulation patterns of metabolites and secondary metabolite accumulation in the leaves of Cyclocarya paliurus at different growth stages. Moreover, studies of secondary metabolic pathways alone have yielded limited information, and this study based on the metabolome and transcriptome comprehensively revealed the differential expression of metabolic regulatory networks across different leaf development stages. This study also characterized the trade-off between primary and secondary metabolism and comprehensive presentation of the overall picture of metabolic pathway.
Overall, manuscript content is interest to the readers. Typographical errors throughout the text requires to be corrected. However, I have some suggestions to improve the presentation of this manuscript.
Major concern is Language. Major revision of the English language is needed.
Minor corrections are as follows
Abstract: This section is short and presents the important findings of this study.
Introduction: This section gives a clear background on essentiality of this study by referring appropriate previously conducted studies.
Line 30, The secondary metabolites of C. paliurus can improve insulin
Line 35, accumulation pattern in C. paliurus leaves has been
Line 36, extensively studied previously.
Line 37, enzyme expression in flavonoids and triterpenoids biosynthetic pathways
Line 38, Appropriate word for fertilization level must be used. Instead of fertilization, level of fertilizer or fertilizer level or fertigation or fertilizer application, etc., throughout the text.
Line 44, network study of C. paliurus
Line 54, to analyze plant economic traits development in fruits
Materials and methods:
Line 63, The annual average temperature is 15.4 ℃,
Line 68-69, and three samples in each
Line 70-71, Leaves with consistent age, similar growth, and no obvious diseases and pests were selected for further analysis.
Line 76-77, Figure 1, Captions are not matching with pictures, It should be A= mature green leaf stage; B = intermediate stage; C = young red leaf stage; D = C. paliurus saplings with different stages of leaf growth.
Line 79-81, Metabolome detection was based on Ultra Performance Liquid chromatography (UPLC) and Tandem mass spectrometry, MS/MS (Applied Biosystems 4500 QTRAP).
Line 81-83, It is not clear, reframe the sentences.
Line 87, Transcriptome data of leaf samples from different growing stages were generated..
Line 94, RNA sequences raw data generated in this study were up loaded at: www.ncbi.nlm.nih.gov/bioproject/PRJNA723183 (Sun et al. 2023).
Line 95, this is then assembled by
Line 99, orthologous group clusters (COG).
Section 2.4, rewrite this section.
Results: Well-presented but exhaustive.
Please include the figures which is essentially required. Other picture can be given in supplementary file.
Captions of all figures must be self-explanatory, please change the captions appropriately. For example Figure 6. O2PLS analysis, readers cannot understand much based on this caption.
Line 129, which ceritificated C.paliurus
Line 166-176, rewrite this section.
Line 178, Based on the association analysis,
Line 196, proline, and lutein
Line 245-247, Naringenin chalcone was considered to be essential substrates of flavonoids, which content decreased significantly along with phenylalanine and 5-O-p-Coumaroylquinic acid. This required to be discussed in the discussion section.
Figures 10-15, it can be combined as single figure (like 10A, 10B, 10C, 10D & 10E) for easy understanding of readers and presented in single page.
Figure 3&4 can be combined as single figure
Line 282-285, phosphomevalonate kinase and diphosphomevalonate decarboxylase in mevalonate pathway of terpenoid backbone biosynthesis has been up regulated resulting in triterpenoids accumulation further (Sun et al. 2022) must be moved to discussion section.
Discussion:
Short and meaningful discussion.
Line 299, Camellia sinensis, must be in italics
Line 301, also found that
Line 317, use appropriate word for fertilization
Line 319-320, Sheng et al. (2021) has found total flavonoids content first increased and then decreased in different growing stage of C. paliurus leaf,
Line 319-323, It is very long sentence and difficult to understand. Reframe the sentence.
Line 323, Use appropriate word for samely?
Line 324, must be C. paliurus
Line 335, C. paliurus, italics
Line 335, plant flavonoids accumulation, and middle nitrogen fertilization level, this is not meaningful to the readers.
Line 351-353, Reframe the sentence.
Line 356, C. paliurus, italics
Line 363, with the idea that the degassing??? of glutamine
Line 365, Use appropriate word for samely?
Line 366, Ganoderma lucidum, italics
Line 367, As a result, amino acids degration??? was considered
Line 368, happened in ordinary??? leaves to
Conclusion part is comprehensive and scientific presentation.

English language revision is needed.
Author Response
Dear editor and reviewer,
I have revised all the tips suggested by reviewer. I want to show my thankful for your patience and work. Looking for good news and wish you a happy day.
Sun Caowen
Title must be “Transcriptomic and non-targeted metabolomic analyses revealed differences in the metabolic networks during leaf coloring of Cyclocarya paliurus”
Thanks, the title has been revised.
Synthetic accumulation of secondary metabolites is a complex process that is largely influenced by the trade-off between primary and secondary metabolism, in addition to being regulated by the synthesis pathway itself. This study successfully demonstrated roles of improved N- assimilation and lower carbon assimilation and the modulation patterns of metabolites and secondary metabolite accumulation in the leaves of Cyclocarya paliurus at different growth stages. Moreover, studies of secondary metabolic pathways alone have yielded limited information, and this study based on the metabolome and transcriptome comprehensively revealed the differential expression of metabolic regulatory networks across different leaf development stages. This study also characterized the trade-off between primary and secondary metabolism and comprehensive presentation of the overall picture of metabolic pathway.
Overall, manuscript content is interest to the readers. Typographical errors throughout the text requires to be corrected. However, I have some suggestions to improve the presentation of this manuscript.
Major concern is Language. Major revision of the English language is needed.
Thanks, English language has been revised.
Minor corrections are as follows
Abstract: This section is short and presents the important findings of this study.
Introduction: This section gives a clear background on essentiality of this study by referring appropriate previously conducted studies.
Line 30, The secondary metabolites of C. paliurus can improve insulin
Thanks, it has been modified.
Line 35, accumulation pattern in C. paliurus leaves has been
Thanks, it has been modified.
Line 36, extensively studied previously.
Thanks, it has been modified.
Line 37, enzyme expression in flavonoids and triterpenoids biosynthetic pathways
Thanks, it has been modified.
Line 38, Appropriate word for fertilization level must be used. Instead of fertilization, level of fertilizer or fertilizer level or fertigation or fertilizer application, etc., throughout the text.
Thanks, it has been replaced by fertilizer level.
Line 44, network study of C. paliurus
Thanks, it has been modified. fertilizer level
Line 54, to analyze plant economic traits development in fruits
Thanks, it has been modified.
Materials and methods:
Line 63, The annual average temperature is 15.4 ℃,
Thanks, it has been modified.
Line 68-69, and three samples in each
Thanks, it has been modified.
Line 70-71, Leaves with consistent age, similar growth, and no obvious diseases and pests were selected for further analysis.
Thanks, it has been modified.
Line 76-77, Figure 1, Captions are not matching with pictures, It should be A= mature green leaf stage; B = intermediate stage; C = young red leaf stage; D = C. paliurus saplings with different stages of leaf growth.
Thanks, it has been modified.
Line 79-81, Metabolome detection was based on Ultra Performance Liquid chromatography (UPLC) and Tandem mass spectrometry, MS/MS (Applied Biosystems 4500 QTRAP).
Thanks, it has been modified.
Line 81-83, It is not clear, reframe the sentences.
Thanks, it has been reframed.
Line 87, Transcriptome data of leaf samples from different growing stages were generated..
Thanks, it has been modified.
Line 94, RNA sequences raw data generated in this study were up loaded at: www.ncbi.nlm.nih.gov/bioproject/PRJNA723183 (Sun et al. 2023).
Thanks, it has been modified.
Line 95, this is then assembled by
Thanks, it has been modified.
Line 99, orthologous group clusters (COG).
Thanks, it has been modified.
Section 2.4, rewrite this section.
Thanks, this section has been rewrited.
Results: Well-presented but exhaustive.
Please include the figures which is essentially required. Other picture can be given in supplementary file.
Thanks, the figures have been simplifying.
Captions of all figures must be self-explanatory, please change the captions appropriately. For example Figure 6. O2PLS analysis, readers cannot understand much based on this caption.
Thanks, the figures captions have been modified appropriately.
Line 129, which ceritificated C.paliurus
Thanks, it has been modified.
Line 166-176, rewrite this section.
Thanks, it has been modified.
Line 178, Based on the association analysis,
Thanks, it has been modified.
Line 196, proline, and lutein
Thanks, it has been modified.
Line 245-247, Naringenin chalcone was considered to be essential substrates of flavonoids, which content decreased significantly along with phenylalanine and 5-O-p-Coumaroylquinic acid. This required to be discussed in the discussion section.
Thanks, the sentence has been rewrited.
Figures 10-15, it can be combined as single figure (like 10A, 10B, 10C, 10D & 10E) for easy understanding of readers and presented in single page.
Thanks, it has been modified.
Figure 3&4 can be combined as single figure
Thanks, it has been modified.
Line 282-285, phosphomevalonate kinase and diphosphomevalonate decarboxylase in mevalonate pathway of terpenoid backbone biosynthesis has been up regulated resulting in triterpenoids accumulation further (Sun et al. 2022) must be moved to discussion section.
Thanks, it has been moved to discussion section.
Discussion:
Short and meaningful discussion.
Line 299, Camellia sinensis, must be in italics
Thanks, it has been modified.
Line 301, also found that
Thanks, it has been modified.
Line 317, use appropriate word for fertilization
Thanks, it has been modified.
Line 319-320, Sheng et al. (2021) has found total flavonoids content first increased and then decreased in different growing stage of C. paliurus leaf,
Thanks, it has been modified.
Line 319-323, It is very long sentence and difficult to understand. Reframe the sentence.
Thanks, the sentence has been reframed.
Line 323, Use appropriate word for samely?
Thanks, the sentence has been modified.
Line 324, must be C. paliurus
Thanks, it has been modified.
Line 335, C. paliurus, italics
Thanks, it has been modified.
Line 335, plant flavonoids accumulation, and middle nitrogen fertilization level, this is not meaningful to the readers.
Thanks, it has been modified.
Line 351-353, Reframe the sentence.
Thanks, the sentence has been reframed.
Line 356, C. paliurus, italics
Thanks, it has been modified.
Line 363, with the idea that the degassing??? of glutamine
Thanks, the sentence has been rewrited.
Line 365, Use appropriate word for samely?
Thanks, it has been modified.
Line 366, Ganoderma lucidum, italics
Thanks, it has been modified.
Line 367, As a result, amino acids degration??? was considered
Thanks, the sentence has been removed.
Line 368, happened in ordinary??? leaves to
Thanks, the sentence has been removed.
Conclusion part is comprehensive and scientific presentation.
Thanks very much.
Less...

Round 2
Reviewer 2 Report
The authors answered my questions well and revised the manuscript accordingly. I don't have any other questions.